# Picometer polar atomic displacements in strontium titanate determined by resonant X-ray diffraction

Carsten Richter [1,2], Matthias Zschornak[1], Dmitri Novikov[2], Erik Mehner[1], Melanie Nentwich[1], Juliane Hanzig[1], Semën Gorfman[3] & Dirk C. Meyer[1]

Physical properties of crystalline materials often manifest themselves as atomic displacements either away from symmetry positions or driven by external fields. Especially the origin of multiferroic or magnetoelectric effects may be hard to ascertain as the related displacements can reach the detection limit. Here we present a resonant X-ray crystal structure analysis technique that shows enhanced sensitivity to minute atomic displacements. It is applied to a recently found crystalline modification of strontium titanate that forms in single crystals under electric field due to oxygen vacancy migration. The phase has demonstrated unexpected properties, including piezoelectricity and pyroelectricity, which can only exist in non-centrosymmetric crystals. Apart from that, the atomic structure has remained elusive and could not be obtained by standard methods. Using resonant X-ray diffraction, we determine atomic displacements with sub-picometer precision and show that the modified structure of strontium titanate corresponds to that of well-known ferroelectrics such as lead titanate.

[1] Institute of Experimental Physics, TU Bergakademie Freiberg, Leipziger Strasse 23, 09596 Freiberg, Germany. [2] Photon Science, Deutsches Elektronen-Synchrotron DESY, Hamburg 22607, Germany. [3] Department of Materials Science and Engineering, Faculty of Engineering, Tel Aviv University, Tel Aviv 69978, Israel. Correspondence and requests for materials should be addressed to C.R. (email: carsten.richter@posteo.de)

Next to high beam intensity, low emittance, and short pulse durations, one of the main benefits of synchrotron light sources is the wide range of accessible photon energies from visible light to hard X-rays. This degree of freedom allows to perform spectroscopy in the X-ray regime (XAS) but also to enhance traditional X-ray scattering methods. This is based on the variation of the complex atomic scattering for photon energies close to electronic transitions in a certain chemical species and therefore referred to as anomalous or resonant X-ray scattering.

Today, the field of resonant X-ray diffraction (RXD) spans a set of techniques based on the different dependencies in the resonant atomic scattering amplitude[1, 2]. The variation in its real and imaginary part is used for contrast enhancement of atoms with similar[3] or even equal[4] number of electrons, or to solve the phase problem in crystallography[5, 6]. Small oscillations of the atomic scattering amplitude beyond absorption edges are used for site-selective study of coordination geometry[7]. Other methods rely on the dependence of the scattering amplitude on the X-ray polarization and wave vector. These make the resonant atom an anisotropic scatterer, which can result in additional forbidden reflections[8] and allows to study different origins of reduced symmetry such as spin, charge order, orbital order, or point defects[9–12].

In this work, we describe a way to enhance the sensitivity of RXD to atomic displacements in the picoscale. In contrast to those above, this our approach employs the energy dependence of atomic scattering amplitudes to vary the structure factor aiming at a suppression of a selected Bragg reflection through the destructive interference of waves scattered by different chemical species. As a result, the structure factor is highly sensitive to a slight dynamic or static displacement of atoms. Moreover, it allows to refine the positions of atoms by analyzing the energy dependence of a few of these reflections. This way, normalization problems are avoided and a structure determination becomes possible in many cases where conventional X-ray crystallography cannot yield this information, e.g., for thin films or heterogeneous samples.

We apply this method to strontium titanate ($SrTiO_3$, STO) both in cubic and polar state. STO receives a lot of attention as a basis for a wide variety of complex oxide electronics. Despite being a wide-bandgap semiconductor with centrosymmetric crystal structure under normal conditions, it can acquire magnetic or ferroelectric properties, become conducting and even superconducting via doping, strain or defect manipulation[13–17]. Furthermore, it was found that single crystals of centrosymmetric STO undergo a structural transition under static external electric field[18]. This phase transition is quite unusual as it is triggered by a redistribution of oxygen vacancies in the crystal. It leads to a distortion of unit cells which is stabilized by the electric field and eventually results in the formation of a transformed layer at the anode side. This modified structure is therefore referred to as migration-induced, field-stabilized, polar (MFP) phase[19] and is connected to the substrate by a strain gradient. A loss of centrosymmetry in the MFP layer was concluded due to the appearance of additional Raman modes as well as piezo- and pyroelectricity[19–21]. On that basis, a transition from space group $Pm\bar{3}m$ to $P4mm$ was predicted[20], but no information on the atomic positions for the MFP phase could be obtained so far. A structure analysis using conventional methods is not feasible due to the restricted geometry and the overlap of reflections from different layers. This problem is avoided using the outlined resonant RXD method. To test it, we characterize the initial, cubic state of STO and obtain the anisotropic mean square atomic displacements (Debye–Waller factors). Based on that, a refinement of atomic positions corresponding to polar displacements in the MFP phase is performed.

## Results

**Resonant RXD near destructive interference.** Although the underlying physics are presented in the Methods section and are common with related RXD techniques, the way we use the resonance in our measurements is different and should be introduced. The central feature is the dependence of the Bragg peak intensity on the photon energy near an absorption edge. The intensity is determined by the norm of the complex structure factor (according to Eqs 1 and 3 in Methods) that runs through a loop in the complex plane when the energy passes the edge (see Fig. 1). This behaviour of the structure factor has been utilized in various anomalous diffraction methods (see Discussion). Beyond that, the Bragg intensity will show dramatic variations in case this loop is close to the complex zero. This leads to a strongly pronounced energy dependence of the intensity which will be particularly prone to structural changes and allow detailed conclusion about absolute values for mean and variance of the atomic positions in a crystal.

In Fig. 1, this situation is sketched for the specific example of the STO 007 reflection near the strontium edge considering two types of positional variation of the titanium atom: a static shift as well as a mean square deviation (due to thermal motion). Only for this illustration, the calculation of the shown curves is based on a simplified model, neglecting the resonant fine structure. As the curves are close to the origin of the complex plane for a certain energy, the measured Bragg intensity as a function of energy will undergo a pronounced minimum. One can see that both considered structural modifications result in orthogonal responses of the progression of the structure factor which can be understood by developing the structure factor for small displacements (see Eq. 5 in Methods). Using the intensity minimum (distance from origin) and its energy position, it is already possible to assess both structural parameters. In a real experiment, the minimum position in energy can be easily determined. The absolute intensity at this position is difficult to measure, but a comparison with several points of the energy dependence can serve as normalization. In addition, the complete curve carries further information, e.g., about the direction of atomic displacement.

However, the obtained results and the high precision would be based on the knowledge of all remaining internal structural parameters. In most cases, there is an uncertainty in more than one atomic position. Even for known structures, the given error margins for position or mean square displacement (Debye–Waller factors) are often too large to give a reliable prediction of the spectral shape near destructive interference. To reach the aspired resolution in atomic positions below 1 pm, this sensitivity and its correlations with static atomic displacements is thus discussed in detail and improved anisotropic atomic displacement parameters (ADPs) are determined before the refinement of atomic positions. Other factors influencing the results, such as site occupancy and valence state, are usually known but can also be disentangled based on the scattering vector dependence of the Debye–Waller factors.

In this work, using several reflections, we vary the weighting of the different atomic scattering amplitudes to gain additional information allowing us to refine all free parameters. We found suitable conditions to perform such measurements for Bragg reflections where two of the Miller indices are even and the other is odd with respect to the pseudocubic setting. If atomic displacement is neglected, the structure factor for this set of reflections is composed of the simple amplitude difference $F \approx f_{Sr} - f_O - f_{Ti}$. Usually, these reflections show a substantial intensity since strontium is the dominating scatterer. Near the strontium $K$ absorption edge, however, the real part of the strontium scattering amplitude is strongly reduced, causing the

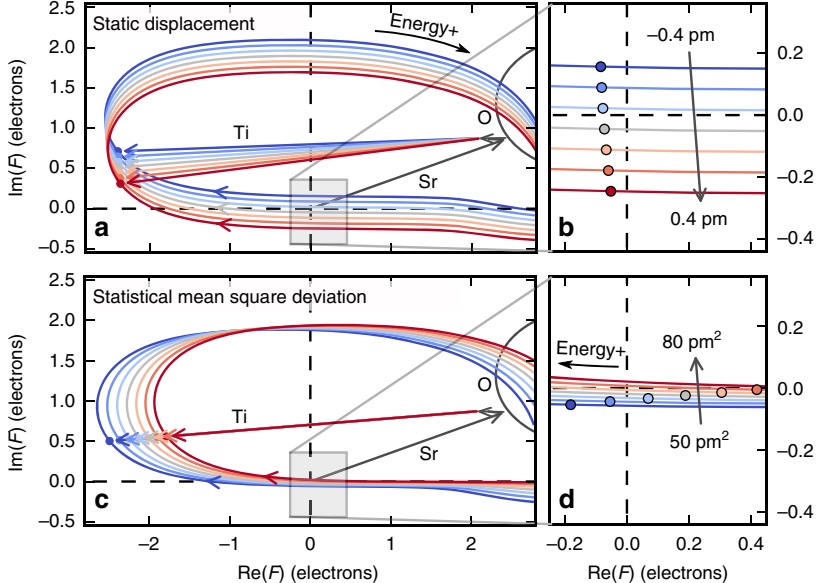

**Fig. 1** Visualization of the methodological approach. Shown are curves described by the complex structure factor $F$ as function of photon energy calculated via Eq. 3 for the specific case of the 007 reflection of strontium titanate in vicinity of the Sr-$K$ edge and neglecting fine structure for clarity. In **a**, **c**, vectors **Sr**, **Ti**, and **O** indicate the atomic contributions to the structure factor for the edge energy of 16,103 eV. The curves are a result of the sum of these vectors and the strong, resonant variation of the strontium contribution. The clockwise direction corresponds to increasing photon energy. Different colors (blue to red) correspond to a variation of mean static displacement (**a**, **b**) or statistical mean square deviation (**c**, **d**) of the titanium atom in the [001] direction. In the zoom on the origin (**b**, **d**), circles mark a selected energy of 16,000 eV corresponding to a structure factor close to the origin where the intensity is minimal. This shows that the energy position of the minimum depends very differently on the two structural distortions

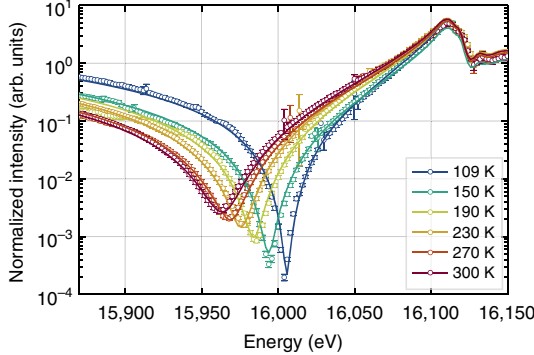

**Fig. 2** Temperature response of bulk $SrTiO_3$. Energy dependencies of the peak intensity of the 007 Bragg reflection near the Sr-$K$ edge are shown for different temperatures. Open circles are the measured values normalized to be on the same scale for each temperature. Error bars indicate the expected SD calculated from the incident and diffracted beam intensity. Solid lines show the fit after varying the atomic displacement parameters based on absorption corrected kinematic diffraction. The distinct shift of the minimum position is a result of changing Debye–Waller factors due to thermal displacement

structure factor to approach zero and leading to the pronounced minimum in intensity.

**Characterization of dynamic thermal displacement**. In order to test the method but also to get a good starting model for the characterization of structural modifications, we characterized the thermally induced atomic displacement for the initial (cubic) state of $SrTiO_3$ where all average atomic positions are known. The experiments involve a measurement of the Bragg intensity as a function of energy while the crystal is kept in the maximum of the

Bragg condition by adjusting the scattering angles resulting in a fixed momentum transfer on the photons.

A high sensitivity of these RXD spectra to thermal displacement is evident in Fig. 2 where the measurement is performed near the Sr-$K$ edge and for different temperatures. A clear trend can be seen and reflects that the Debye–Waller factors of the different atoms do not scale proportionally with temperature. This fairly corresponds to the changes initially calculated (Fig. 1, bottom) for increasing mean square displacement. The simulated curves in Fig. 2 are based on Eq. 1 and the refinement of the anisotropic ADPs $U^{ij}$ for each atom that determine the Debye–Waller factors (see Methods). For energies beyond the edge at $\approx 16,100$ eV, it was necessary to include an absorption correction and the resonant fine structure.

In cubic $SrTiO_3$, oxygen occupies a noncubic Wyckoff position leading to two different oxygen ADPs necessary to describe the measured intensity correctly—one parallel and one perpendicular to the Ti–O bond. The results of least-squares refinement displayed in Fig. 2 were based on the assumption that the ADPs show a temperature dependence according to the Debye model[22, 23] and taking into account a decomposition of the Debye temperature $\Theta$ for oxygen into a parallel and a perpendicular part following Potzel et al.[24]. For this reason, it was necessary to record spectra of additional reflections to ensure an unambiguous refinement of the ADPs. Table 1 summarizes the results for anisotropic ADPs and Debye temperatures obtained by including room temperature measurements of the reflections 005, 205, 405, 504, and 016 in a similar energy region as for the 007 reflection (see Supplementary Fig. 3).

Despite its lower mass, titanium shows a smaller thermal vibration than strontium and the oxygen oscillations along the Ti–O bond are much smaller than perpendicular to it. Both observations are in good agreement with literature results[25] and the absolute values are within the error margins. This is remarkable as the results were obtained by different methods. The errors shown in Table 1 seem relatively high even though the

spectra are very sensitive to small variations in the results. A detailed error analysis reveals that the confidence volumes are comparatively small and that the large uncertainties originate from a strong correlation between the parameters. There is a much higher certainty in relative values of the parameters as the confidence regions are only elongated along a direction, which corresponds to proportional change of the Debye temperatures (illustrated in Supplementary Fig. 4). Therefore, we assume that the ratios of values given in Table 1 are more accurate than those taken from literature. To get more precise absolute values, the correlation of fit parameters can in general be further reduced by including spectra of additional reflections.

**Static displacement and the structure of the MFP phase.** Based on the refined ADPs of cubic SrTiO₃, it is possible to characterize the structural changes in the strained MFP layer. The MFP phase is formed under influence of a [001] oriented, static electric field of about 1 MV m⁻¹ in a few micrometer thick layer on the anode side of the crystal (see sketch in Fig. 3) where the oxygen vacancies are depleted. It can be observed by an increase of the lattice parameter along the field direction of up to 1%. A tetragonal distortion with the unique axis $c$ in field direction was found with a Poisson's ratio of approximately $\nu = -\frac{\Delta a}{\Delta c} = 0.2$[19] which conforms to literature values[26]. Although the change of lattice parameters alone does not modify the structure factor, the associated atomic displacements should have a clear effect on the energy dependence of the intensity. The formation of the MFP phase is observed as an increased current through the sample, which reaches its maximum after a few hours and is attributed to the migration of oxygen vacancies. It is accompanied by a broadening of Bragg peaks and formation of pronounced

shoulders on their lower-angle sides. Both the current and the broadening converge after a period of about 8 h depending on sample and ambient conditions. Following symmetry considerations, a displacement of atoms perpendicular to the field direction would result either in an orthorhombic structure and a consequential twinning/peak splitting or in a two-fold superlattice in $a$–$b$ direction[27] causing additional superlattice reflections. We recorded reciprocal space maps and ruled out both cases concluding that all atomic displacement must be parallel to the field. In addition, we defined the position of the strontium atom as the origin of the unit cell and assumed that the MFP phase exhibits the same ADPs, as they were found above for cubic SrTiO₃, which is justified due to the very small static displacements in the structure. Therefore, only three free parameters are left which are the $z$ positions of titanium and the two oxygen sites that are not anymore linked by symmetry: for one the Ti–O bond is parallel and for the other it is nearly perpendicular to the electric field vector.

As its strain in equilibrium depends on the applied electric field strength, crystal thickness, as well as sample history, there is no prototype of the MFP phase. Furthermore, there is a continuous decrease of strain towards the cubic bulk SrTiO₃, which can be seen in the form of a plateau between the two corresponding Bragg peaks. To study differently strained volumes in an isolated manner, we keep the momentum transfer **Q** fixed at the corresponding position. In particular, in the conditions of our experiments, we observed a distinct maximum in the diffraction profile towards smaller momentum transfer (see Fig. 3b) corresponding to a value of $-\frac{\Delta Q_z}{Q_z} = \frac{\Delta c}{c} = 0.15\%$. This maximum is related to a region of constant strain associated with the MFP phase near the crystal surface. Therefore, we select the MFP by choosing accordingly the value of $Q_z$ ($l$) for each studied reflection $hkl$. At these positions of reciprocal space, we then measured the energy dependence of Bragg intensity for several reflections in a similar region as before the electroforming. The resulting data along with calculations are shown in Fig. 4.

As it is on the order of the penetration depth of the X-rays, the refinement also includes the thickness of the probed, strained region using Eq. 2 from the Methods part. It does not affect the main feature (minimum intensity and position) but is necessary to obtain a good fit for energies beyond the edge. For small thicknesses, the influence of a covering layer on the absorption correction is weaker and was not taken into account. Depending on the reflection, the kinematic penetration depth varies from 18 to 43 μm below and from 5 to 10 μm beyond the absorption edge. Extinction effects have also been neglected, as the measured

### Table 1 Refined atomic displacement parameters for cubic SrTiO₃

| Atom | Debye temperature $\Theta$ (K) | ADP at 298 K $U$ (Å²) |
|---|---|---|
| Sr | $198^{+31}_{-43}$ | $0.0127^{+0.0080}_{-0.0031}$ |
| Ti | $292^{+56}_{-73}$ | $0.0109^{+0.0084}_{-0.0032}$ |
| $O_\parallel$ | $713^{+364}_{-270}$ | $0.0061^{+0.0085}_{-0.0030}$ |
| $O_\perp$ | $499^{+92}_{-121}$ | $0.0117^{+0.0082}_{-0.0031}$ |

Results of fitting the temperature dependent experimental RXD spectra to the Debye model. The given errors correspond to 1σ uncertainties

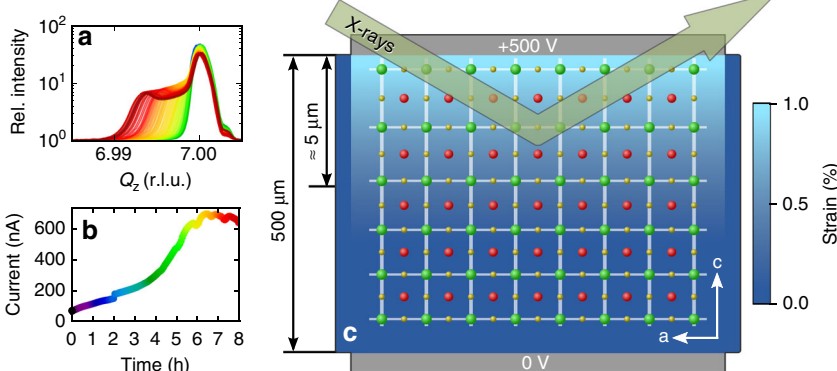

**Fig. 3** Characterization during formation of the MFP phase The [001] oriented SrTiO₃ single crystal is sandwiched between two titanium electrodes such that the electric field is directed perpendicular to the surface (**c**, not drawn to scale). A few micrometer thick, strained MFP layer can be observed only on the anode side where the X-ray measurements are performed. The diffraction profiles (**a**, here 007 reflection) show a shoulder developing with at lower out of plane momentum transfer $Q_z$ (in terms of reciprocal lattice units–r.l.u.) while the leakage current through the crystal (**b**) increases

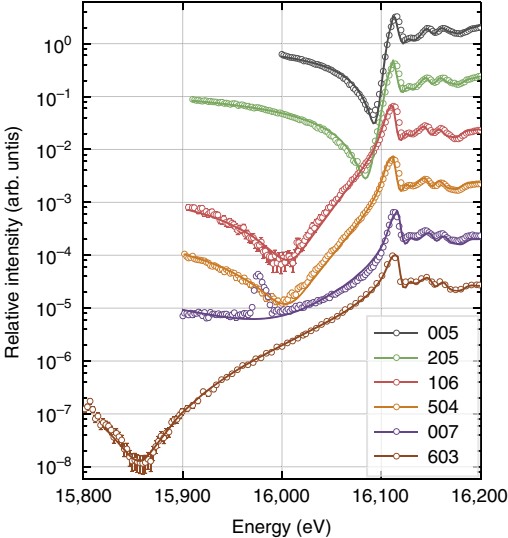

**Fig. 4** Energy dependence for several Bragg reflections of the MFP phase. The Bragg intensity (open circles) was measured selectively for the strained MFP layer formed on the SrTiO₃ substrate. Solid lines show the calculation after refinement of *z*-positions of each atom in the unit cell. Error bars indicate the standard deviation calculated from the incident and diffracted beam intensity. For better visibility, the spectra have successively been scaled by a factor of 10 in the same order as they appear in the legend. The energy dependence of the 007 reflection differs strongly from the one acquired at room temperature before formation (compare Fig. 2). This clearly indicates a phase transition since lattice constants do not influence the energy dependencies

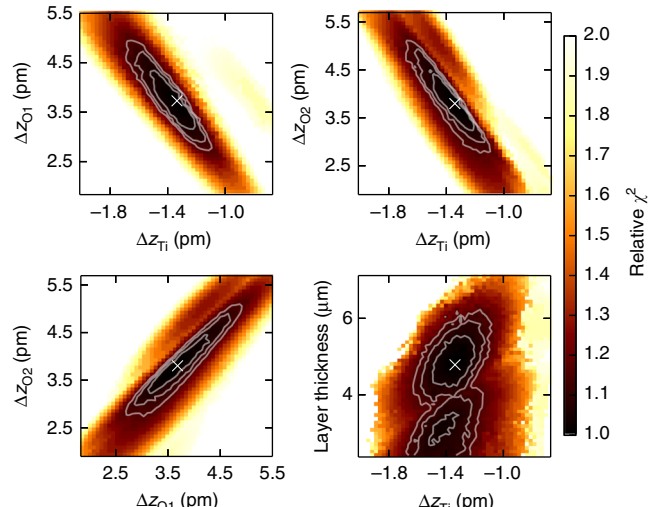

**Fig. 5** Confidence intervals of the fitted displacement parameters. Correlations are illustrated by the dependence of the residual sum of squares $\chi^2$ on pairwise variation of the fit parameters (atomic displacements) during fit of the data shown in Fig. 4. The white cross indicates the global minimum. Confidence regions for 1σ, 2σ, and 3σ are marked by white contour lines following Avni et al.[51] Regarding the layer thickness, only correlation with titanium displacement is shown. The uncertainty of the thickness parameter has a negligible effect on atomic displacements

### Table 2 The structure of the MFP phase refined in space group *P4mm*

| Atom | Wyckoff site | *x/a* | *y/a* | *z/c* | Δz (pm) |
|------|-------------|-------|-------|-------|---------|
| Sr | 1*a* | 0 | 0 | 0 | – |
| Ti | 1*b* | 1/2 | 1/2 | 0.4966 (3) | −1.34 (16) |
| O1 | 1*b* | 1/2 | 1/2 | 0.0096 (11) | 3.67 (59) |
| O2 | 2*c* | 1/2 | 0 | 0.5099 (11) | 3.80 (63) |

The result refers to a selected strain of $\frac{\Delta c}{c} = 0.15\%$ (c = 3.911 and a = 3.904 according to the Poisson's ratio). The origin was defined at the Sr position. Given errors correspond to 1σ uncertainty. The refinement includes measurements of spectra for six reflections (see Fig. 4)

reflections are typically weak. Furthermore, we assumed that the fine structure does not change significantly during phase transition to the MFP phase. This approximation seems to be valid as the fine structure oscillations (see Fig. 4 beyond 16,100 eV) are well modeled using spectra obtained from bulk SrTiO₃. The validity is further discussed with aid of absorption spectroscopy measurements in Supplementary Note 1.

The results of the refinement of internal structure parameters are summarized in Table 2. They confirm that the MFP structure is polar. It belongs to the same structure type as ferroelectric, tetragonal BaTiO₃ and PbTiO₃ with space group *P4mm*. In comparison, the absolute displacements of atoms from high symmetry positions are significantly smaller in the present case of the MFP phase. However, similar to PbTiO₃, the displacement of titanium is roughly a factor of three smaller than those of oxygen. The relative uncertainties in the refined results are much smaller than in the case of Debye temperatures. Even though the refinement was based on iterating through all combinations of ADPs within the confidence volume, the error did not propagate proportionally indicating a stronger sensitivity to the rather small

static displacement. Figure 5 illustrates the pairwise correlations of all displacement parameters. The derived error margins correspond to a proportional movement of the considered atoms in *z*-direction and away from their original high-symmetry positions meaning that the ratio of the shifts is again known more precisely than one would estimate based on Table 2. The layer thickness, in turn, shows minor correlation and a less well defined confidence region. It should also be pointed out that, using the described method, the accuracy in the determination of average atomic positions is at the sub-picometer level.

**Strain dependence of atomic displacement.** The MFP phase is connected with the bulk SrTiO₃ crystal via a strain gradient. An open question is, how the displacement evolves with strain or whether a polar phase is only formed in the thin layer of constant strain closest to the anode which is associated with the additional peak in the diffraction profile. In the latter case, the region of continuous strain gradient, which is seen as a plateau between the peaks (see Figs 3 and 6), would be assumed to be tetragonally distorted but to remain centrosymmetric like the substrate. To answer this question, we performed similar energy scans for several fixed values of the momentum transfer **Q** selecting the MFP phase, the substrate and also the transition region but restricting ourselves to the 007 and 205 Bragg reflections. Although significant uncertainties are expected in a refinement of atomic positions that relies on only two reflections, a loss of centrosymmetry would have an unequivocal effect on the measured spectra. Assuming a proportional movement of all atoms, we further reduced the amount of parameters and can give an estimate of all atomic positions based on the former results in Table 2.

Regions of the sample having different strain in *c* direction were selected using four positions of the $Q_z$ scans (see Fig. 6) near the bulk reflections 007 and 205. One should note that the model of a strain gradient predicts that there is no region of constant strain with a finite width. From the experimental conditions, one

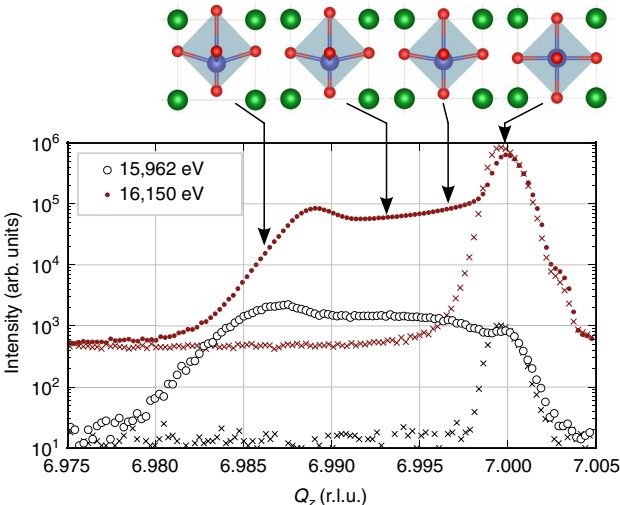

**Fig. 6** The 007 reflection profiles of SrTiO₃ before and after formation of the MFP phase. The maximum on the right hand side corresponds to cubic (bulk) SrTiO₃ and is the only feature under normal conditions (crosses). After formation (circles), a shoulder peak is observed at lower momentum transfer $Q_z$, which is associated with the MFP phase. The intensity ratio of both peaks is very different when measured at two different energies near the Sr-K edge –15,962 eV (black) and 16,150 eV (red, shifted to top). This can be explained by a change of the relative, crystallographic positions of the atoms. These atomic displacements were obtained from fitting the energy dependencies (see Fig. 7). The unit cells in the top of the figure (green: strontium, blue: titanium and red: oxygen) show the displacements enhanced by a factor of 10 for better visibility. Arrows mark values for which fixed **Q** energy scans have been recorded

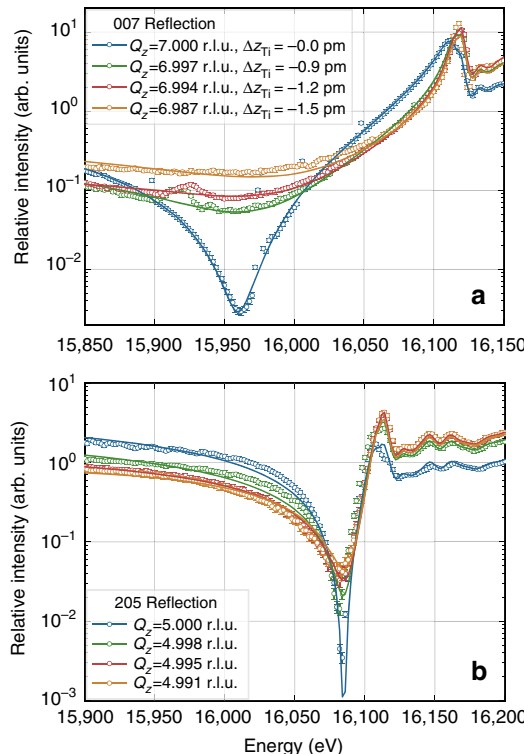

**Fig. 7** Response of RXD curves to strain induced displacement in the MFP phase. Energy dependencies of the 007 (**a**) and 205 (**b**) reflections are shown for different positions on the diffraction profile along $Q_z$, including both MFP and cubic phase of SrTiO₃, as indicated in the legend. Each selected value of $Q_z$ corresponds to a differently strained region in the sample given in terms of reciprocal lattice units (r.l.u.) with respect to the cubic state. The solid lines represent the best fit after coupled variation of the z-positions of the atoms (see text) and error bars indicate the SD calculated from the incident and diffracted beam intensity. The corresponding results for titanium displacement are given in the legend

can estimate how thick the layers are that contribute to a certain position in $Q_z$ scans in reciprocal space. This assessment is based on the value of the strain gradient. In an exponential strain profile model for our case, we obtained an upper limit for the strain gradient of $1.5 \cdot 10^{-3}$ μm⁻¹ (near the surface) corresponding to a value of $5.9 \cdot 10^{-7}$ per unit cell. For the 007 reflection, this would result in a $\frac{\pi}{2}$ phase shift after 350 unit cells and therefore in a thickness of 140 nm of the contributing layer. Based on this thickness, the calculated peak width for the 007 reflection is $\Delta Q_z \approx 0.0035$ r.l.u. (reciprocal lattice units). This estimation has been done for the position of highest strain. For lower strain, the broadening of Bragg peaks will decrease and the depth resolution of the measurement will improve. Therefore, the sampling we used in this work was always below the limits of resolution.

The fixed **Q** energy scans corresponding to the four selected strain values are presented in Fig. 7 and clearly show that also the transition region exhibits a polar structure. Already for the smallest selected value of $\Delta Q_z/Q_z = -0.04\%$ (at $Q_z$ values 6.997 and 4.998 r.l.u.), the energy dependence shows an enormous response which is enhanced with larger strain. The changes in RXD curves saturate when further increasing the strain and this saturation is reached earlier for the 205 reflection. The legend in Fig. 7 lists the titanium displacements that result from the fit with the condition of a proportional movement of the oxygen atoms according to Table 2. These movements (enhanced by a factor of 10) are illustrated in Fig. 6 together with the $Q_z$-scans near the 007 reflection for two selected energies.

The strong response seen in Fig. 7 already for small strain or small displacements suggests that even much smaller perturbations in the average structure can be detected this way. A clear trend of larger atomic displacement and therefore stronger electric polarization with higher strain is observed. Moreover, this

dependence is not linear. The obtained relationship between Ti-displacements and the lattice strain suggests that the displacement plays the role of the primary order parameter in the MFP phase. The lattice strain must be summed up from purely microscopic and purely macroscopic contributions. The microscopic contribution expresses the need to optimize the unit cell size to the new positions of atoms, whereas the macroscopic contribution accounts for the need to connect the strained phase to the nonpolar bulk. We have found that the second contribution has negligible effect on the positions of the Ti-atom. The same relationship between atomic positions and lattice parameters was documented in the piezoelectric crystals under quickly rising electric fields[28, 29], where the lattice strain was caused by atomic displacements due to electric field and elastic vibrations due to the short rising time.

**Comparison with density functional theory calculations.** Electronic structure calculations using density functional theory (DFT) were performed to validate the relation between strain and polar displacements, as well as to estimate the macroscopic polarization resulting from these displacements. The effects caused by the external electric field have been emulated by imposing relative atomic displacements $\Delta z/c$ of titanium and oxygen in field direction as they were experimentally found for the MFP phase according to Table 2. This was followed by geometrical relaxation of the MFP unit cell size. Calculated changes in lattice parameters as well as strain parameters and polarization

**Table 3 Lattice parameters of the MFP phase obtained by means of DFT**

|     | $a$ (Å) | $c$ (Å) | $\Delta a/a$ | $\Delta c/c$ | $\nu$ | $\Delta p_{ion}$ (eÅ) | $\Delta p_{elec}$ (eÅ) | $\Delta p_{tot}$ (eÅ) |
|-----|---------|---------|--------------|--------------|-------|-----------------------|------------------------|-----------------------|
| STO | 3.947261 | 3.947261 | – | – | – | – | – | – |
| MFP | 3.945461 | 3.955710 | −0.0456% | 0.2136% | 0.214 | −0.508 | 0.940 | 0.432 |

A polar distortion was imposed by the relative atomic displacements found in the experiment (see Table 2). Strain parameters including Poisson's ratio $\nu$, as well as electronic ($\Delta p_{elec}$) and ionic ($\Delta p_{ion}$) dipole moment per unit cell caused by the atomic displacements are presented

$\Delta p_{tot}$ are given in Table 3. The projector-augmented wave method[30] in Perdew-Burke-Ernzerhof (PBE) parameterization[31] was employed as implemented in the VASP code[32]. Total energies have converged better than $10^{-10}$ eV with a maximum kinetic energy of 450 eV for the plane-wave basis set and Γ-centered $12 \times 12 \times 12$ Monkhorst–Pack $k$-point meshes with spacings $< 0.02 \times 2\pi\text{Å}^{-1}$. For the evaluation of strain, we kept relative atomic positions fixed and relaxed the cell geometry within the space group to forces less than $10^{-5}$ eV Å$^{-1}$. The change in polarization has been assessed within the framework of a Berry-phase treatment[33].

During the measurements on the structure of the MFP phase, we selected a strain value in $c$-direction of $\frac{\Delta c}{c} = 0.15\%$. This value is in reasonable agreement with the result obtained here by DFT calculations $\frac{\Delta c}{c} = 0.21\%$. The Poisson's ratio even better matches previously observed values for the MFP phase[19].

## Discussion

In this study we obtained highly accurate values for atomic positions of the recently discovered MFP phase of SrTiO$_3$ and presented a resonant RXD method to obtain these even for inhomogeneous samples. The method involves adjusting the X-ray wavelength—and thus the scattering amplitude of the resonant atoms—to tune destructive interference of the partial waves from different scatterers. To which extent this can be achieved strongly depends on the atomic positions, giving an access to reveal slight displacements of the atoms. In the present case, a sensitivity to changes of average positions at the level of 1 pm and below has been demonstrated. Prerequisite for a careful analysis of the data was a refinement of temperature induced atomic displacement parameters of cubic SrTiO$_3$. We could confirm directly that SrTiO$_3$ forms a polar structure under electric field, which is of the same type as ferroelectric BaTiO$_3$ at room temperature. This is remarkable, as, in contrast to BaTiO$_3$ and PbTiO$_3$, SrTiO$_3$ does not form this phase when cooling to low temperatures: the beginning of a paraelectric to ferroelectric phase transition was observed at 40 K but remains incomplete down to 0 K[34]. Yet, it is known that other external influences, such as strain, can stabilize this ferroelectric phase even at room temperature[15]. Using DFT calculations, we could validate the relation between strain and atomic displacement that was experimentally found for the MFP structure and furthermore give an estimate for the resulting electric polarization of the unit cell. We have shown that the magnitude of polar displacement exhibits a nonlinear relation to the strain. The recently found indications for an electrostrictive origin of piezoelectricity in the MFP phase[21] would suggest a quadratic dependence of the strain on polarization. In our case, however, the polarization dependence of strain appears to follow an even higher power law (see legend in Fig. 7), which can be attributed to the elastic interaction of MFP layer and substrate. Therefore, this data adds to a better understanding of the mechanism of formation and the role of the substrate.

The employed resonant suppression of Bragg reflections can be universally combined with all diffraction based X-ray techniques as long as the photon energy of the X-ray source is tunable. There are similar attempts to extract structural information reported in literature, all subject to substantial limitations compared to our development. To avoid ambiguities in the refinement, fixed structural models were presupposed or only one structural parameter was assumed to be unknown[35, 36]. Often, the analysis required inverting (switching) the polarity of the structure[37, 38] with an electric field in order to measure the Friedel pair contrast. This is not always feasible or may alter the material as in the present case. Furthermore, in all cases the impact of Debye-Waller factors was neglected and the obtained resolution was much lower than in our approach, which exploits destructive interference.

In Supplementary Note 2, we briefly compare the method to the variety of other techniques and show that it can be useful for a large group of materials especially in a restricted geometry where only few reflections can be accessed or their absolute intensities cannot be compared, and therefore standard diffraction methods cannot be applied. It also allows to study dynamics in crystals in situ, such as switching in ferroelectric films or polarization changes in pyro- and piezoelectrics and represents a uniquely sensitive tool for cross-checking existing structure models or to reveal pseudosymmetries. Advantages are that the measurements can be performed in the pre-edge region allowing straightforward calculations and that the kinematic approximation of diffraction, neglecting extinction effects, is valid for this method, as it is based on weak reflections.

## Methods

**Theory**. Within the kinematic theory of diffraction from crystals, in the absence of extinction effects, the dependence of the diffracted intensity of a Bragg reflection on the photon energy $E$ is given by[39]

$$I_{kin} \propto A(E)|F(E,\mathbf{Q})|^2 \qquad (1)$$

where $F$ denotes the structure factor and $A$ describes absorption. Further dependencies on $E$, such as in Lorentz and polarization correction, are usually weak and can be approximated as linear. The momentum transfer vector $\mathbf{Q}$ defines the Bragg reflection. In Bragg geometry, the absorption factor can be written as[40]

$$A = \frac{1}{\mu\left(1 + \frac{\sin\alpha}{\sin\beta}\right)}\left[1 - e^{-\left(\frac{1}{\sin\alpha} + \frac{1}{\sin\beta}\right)\mu t}\right] \qquad (2)$$

with $t$ being the thickness of the probed crystalline layer, $\mu$ the linear absorption coefficient, and $\alpha$ and $\beta$ respectively referring to the angle of the incidence and exit of the X-ray beam with respect to the crystal surface. In the independent-atom model, the structure factor is expressed as a weighted sum of the scattering amplitudes $f_j$ of all atoms present in the crystallographic unit cell. Real crystals exhibit a certain degree of disorder due to lattice vibrations and defects. Therefore, an averaging over all unit cells is necessary, which is commonly parameterized with the occupancy of the crystallographic site $o_j$ and the Debye-Waller factor $e^{-M_j}$ taking account of the reduced scattering amplitude due to the uncertainty in the position $\mathbf{r}$ of atom $j$:

$$F(E,\mathbf{Q}) = \sum_{j \in u.c.} o_j f_j(E,\mathbf{Q}) e^{-M_j} e^{i\mathbf{Q}\mathbf{r}_j}. \qquad (3)$$

The Debye–Waller factor is a result of harmonic approximation of lattice vibrations due to temperature and is related to the mean square of the atomic displacement $\mathbf{u}_j$ of the atom $j$ in the direction of $\mathbf{Q}$ via $M_j = \frac{1}{2}\langle(\mathbf{Q}\cdot\mathbf{u}_j)^2\rangle$. The anisotropic mean square displacement is commonly expressed in form of the ADP tensor $U^{ij}$[41]. Neglecting the polarization and wave-vector dependence of the

dispersion corrections $f'$ and $f''$, the scattering amplitude can be written as

$$f(E, Q) = f_0(\mathbf{Q}) + f'(E) + if''(E). \quad (4)$$

In resonant diffraction, it is common to investigate Bragg reflections where the structure factor is zero due to destructive interference of scattered waves coming from atoms of the same species. Under resonant conditions, these atoms may have different scattering amplitudes due to their different chemical or magnetic environment lifting the destructive interference and resulting in non-zero intensity and therefore new, previously forbidden reflections. In this work, we went the opposite way and tuned the scattering amplitude of distinct elements to cancel out each other. This can only be true in a small range of photon energy, since the scattering amplitude of each element shows a different energy dependence. It also strongly depends on any contribution to the structure factor such as occupancy, shifts in the atomic positions, temperature-induced dynamic displacement, defects, or valence state.

Let $\mathbf{u}$ be the small displacement of a selected atom $j$ from the ideal position $(\mathbf{r}_j \rightarrow \mathbf{r}_j^0 + \mathbf{u}; \mathbf{u} = \mathbf{u}_j$ to improve legibility). To point out the contributions of displacement, the structure factor can be approximated, assuming it is close to zero, as

$$F = \underbrace{F|_{\mathbf{u}=0}}_{\rightarrow 0} + \frac{\partial F}{\partial u^k}\Big|_{\mathbf{u}=0} \langle u^k \rangle + \frac{1}{2}\frac{\partial^2 F}{\partial u^k \partial u^l}\Big|_{\mathbf{u}=0} \langle u^k u^l \rangle + \cdots$$
$$\approx o_j f_j e^{-M_j} e^{i\mathbf{Q}\mathbf{r}_j^0} \left( iQ_k \langle u^k \rangle - \frac{1}{2} Q_k Q_l \langle u^k u^l \rangle \right). \quad (5)$$

One can see that the term of the mean displacement carries the imaginary unit resulting in the orthogonal displacement of the complex structure factor for the two contributions that was seen in Fig. 1. It explains why, in the present case, the intensity minimum is shifted in energy due to thermal vibration but only less pronounced in the case of directed displacement of titanium. Furthermore, the measured intensity will be proportional to the square of mean displacement as well as of mean square displacement of the atom, weighted with first and second power of the momentum transfer, respectively. This illustrates an enhanced sensitivity of higher order reflections.

**Experiment and data treatment**. We found that the Bragg intensity has a minimum in the pre-edge region of the Sr-$K$ edge (at 16,150 eV) for a set of reflections where exactly one Miller index is odd. On the other hand, the post-edge region contains valuable information about the direction of a considered atomic displacement since a significant imaginary part is typically added to the scattering amplitude. Owing to variations in the $\mathbf{Q}$-dependent scattering amplitudes, the minimum is located at different energies for different reflections which defines the energy range for each measurement. Energy-dependent RXD data have been collected in quasi-symmetric geometry (i.e., $\alpha = \beta$) at the beamlines BM20[42] and BM28[43] of the European Synchrotron Radiation Facility (ESRF) using the diffractometers in four-circle geometry. The preparation of samples (crystal plus electrodes) has been described in detail elsewhere[19]. To facilitate the formation of the MFP phase, the photon flux had to be reduced such that the leakage current was about 1 µA or lower depending on the formation state. For the study of influences from thermal displacement, a continuous-flow cryostat was used at BM20 to cool the crystals down to a minimum temperature just above the cubic to tetragonal phase transition at 105 K[44].

The fit of the energy dependent Bragg intensities was based on Eq. 1 multiplied by a linear device function where a free scale parameter was included for each reflection. Therefore, only relative intensity changes were relevant for the fit results. The dispersion terms $f'$, $f''$ far off the edge have been taken from the Sasaki database[45], whereas the near edge fine structure in $f'$, $f''$ has been obtained for strontium by means of XAS measurements on SrTiO$_3$ powder at the Sr-$K$ edge and subsequent application of Kramers-Kronig relations following Lucarini et al.[46]. The nonresonant atomic scattering factors $f_0(\mathbf{Q})$ have been taken from the International Tables for Crystallography[47]. Additional peaks appearing in the energy dependent RXD scans are attributed to Umweganregung[48] and have been filtered out through multiple scans for small rotations about the Bragg vector $\mathbf{Q}$ where feasible.

**Data availability**. Raw data were generated at the ESRF and can be accessed from the Open Science Framework[49]. Software routines used for data modeling are available online[50].

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

## Acknowledgements
We thank the teams of beamlines BM20 (ROBL) and BM28 (XMaS) of the ESRF, particularly S.D. Brown and C. Bähtz, for their outstanding support during the measurements. Furthermore, we are very grateful for advice given by V.E. Dmitrienko regarding the first principles calculations and for the help of H. Choe and B. Khanbabaee during one of the beamtimes. Financial support by the German Federal Ministry of Education and Research (BMBF) within the projects 05K13OF1 (C.R.), 03EK3029A (M. Z., E.M., J.H., and M.N.), and 05K13PSA (S.G.) is greatly acknowledged.

## Author contributions
C.R. developed the technique, led the measurements, analyzed the data, performed the database survey, and prepared the initial manuscript. All authors contributed significantly to the manuscript. M.Z. carried out the first principles calculations. C.R., D.N., E.M., M.N., M.Z., and S.G. participated in the experiments. J.H. prepared and pre-characterized the samples. E.M. designed the sample holder with cabling for high-voltage experiments. S.G. contributed to crystallographic considerations prior to the measurements. D.C.M. and M.Z. suggested an investigation of the MFP phase using resonant XRD.

## Additional information

**Competing interests:** The authors declare no competing financial interests.

