## [Peer Review File · Nature Communications]

Reviewers' comments:

Reviewer #1 (Remarks to the Author):

This paper presents interesting results on the structure of SrTiO₃ when it has been exposed to high electric field for some time, which results in an oxygen deficient phase. The paper presents high precision measurements of the positions of the ions in this phase that are obtained through a novel resonant x-ray scattering technique. Overall, I consider this work appropriate for publication in Nature Communications, but there are ways in which this paper should be improved.

My most serious concern is that I do not feel that the novel method is presented in a very clear manner, and certainly not one that a non-specialist can follow.

If I understand correctly the authors first determined the Debye-Waller factors for cubic SrTiO₃. No discussion was presented on whether these represented an improvement over values obtained previously in the literature. The reader is left wondering whether this part of the work is due to the precision of the improved results or not.

There is a similar sense to the presentation of the refinement of the MFP phase structure. While the refinements are good and produce low errors what I would have liked to see is some real attempt to show how the method used is at the core of the success of the experiment. For example, if only a single energy was used how much less precise would the obtained result be?

As the authors are trying to convince the scientific community of a new method, rather than simply presenting an interesting new result I think they need to present a much clearer explanation of the method along with a real assessment of the benefits of their method before this paper can be published.

Reviewer #2 (Remarks to the Author):

This paper reports a resonant X-ray study of the formation of a new crystalline (MFP) phase of SrTiO₃ under electric field with the goal being to clearly expose the atomic displacement in this restricted geometry system.

From the perspective of the science studied in this work:

I am not an expert in resonant X-ray measurements, SrTiO₃ or piezoelectricity, but I don't think solving this problem of the phase of SrTiO₃ is a burning question -- a quick check of the literature on the MFP phase indicates that previous work on this is not particularly highly cited and does not seem to have taken off much beyond the immediate efforts of a subset of the present authors.

From the perspective of a measurement achievement:

The experiments themselves seemed compelling, although the details were hard to follow. There is a huge jump in depth of language/discussion/etc moving from the Introduction to the first Section of the paper. While it seems that all technical information is there, the relationship between different aspects and the flow does not seem optimal. Just to take one example: the authors state that a proper understanding/quantification of the Debye-Waller factor on the resonance for bulk SrTiO₃ is necessary to expose the changes in the film geometries, but the rationale for that is not stated in adequately simple physical terms. A similar comment could be made about the issue of attenuation.

Perhaps more importantly, I would have thought that for demonstrating the merit of a new method (or a variation), a more conventional system which is well (if not fully) understood would be more suitable than considering a problem (the MFP phase) that is open.

Finally, considering other Nature Communication papers I have read, I would argue that the present paper is not written to the standard of readability for a sufficiently broad audience that I would expect for Nature Communications.

Given what I perceive to be a niche problem (at the present time) and the overwhelmingly technical tone of the paper, I would recommend that the authors submit their work in a more specialized journal such as Physical Review B.

I therefore recommend that this paper be rejected from Nature Communications.

Reviewer #3 (Remarks to the Author):

The manuscript describes a resonant X-ray technique to extract polar displacements in the MFP phase of SrTiO₃. The formation of the MFP was described in earlier publications.

The technique is relatively new: others have used similar techniques to extract displacements in perovskites [Phys. Rev. Lett. 105, 217601 (2010); J. Appl. Crystallogr. 37, 193 (2004) -- these should be cited], but did not analyze the diffraction anomalous fine structure energy spectrum. The method used by the authors appears to be robust, with appropriate detail given to error analysis, and the conclusions appear valid.

However, this reviewer feels that the technique of "resonantly suppressed X-ray diffraction" will be of very limited interest, due to the need for a synchrotron, the time necessary to measure the data (and for analysis), the specificity of the method (small atomic displacements), and the fact that other crystallographic techniques (e.g., phase retrieval techniques [Nat. Mater. 2, 99 (2002); J. Phys. Condens. Matter 20, 445006 (2008); etc.]) can be used to extract similar position information, although with not quite the resolution, but over the full thickness of the unknown layer. Publication in a more specialized journal would be appropriate.

Reviewer #4 (Remarks to the Author):

This is an interesting paper that reports on picometer atomic displacements as a result of the application of an electric field on SrTiO₃. The goal and interest of this study has been clearly identified in the abstract and in the introduction, and I do not want to copy and paste the overall originality of this work, which I fully agree with. It is important to characterise these distortions, the method that the authors have used is well suited (see more below) and this reviewer believes that the manuscript should be published in Nature Communications, albeit not in its present form.

Going further into the details of the manuscript, I have the following remarks:

- The wording of the title "...by resonantly suppressed X-ray diffraction" is not appropriate. This sentence is not correct and turns out to be meaningless without reading the paper, which is not the primary goal of a title. Authors should seek for a better title.

- I disagree with the authors statement that they "have developed a novel method of X-ray structure analysis". Fitting the energy profiles to account for crystallographic information is currently done and there are examples in the literature. Reference 10 includes some "old" examples.

- The authors have been very vague about the details of their method and the basic underlying idea is missing. Authors should capture in simple words the main features of their methodology. I agree that the method is embedded in Equation 3. However it is not convincing and the reader has to take authors' word for granted. The characteristic profiles measured and shown in Figures 1, 3, 5 and S1 result from the interference of a rapidly varying complex function, the resonant scattering factor, with a smoothly decreasing non resonant (Thomson) scattering. This kind of profiles stands not only for tabulated f' and f'' but for a variety of complex functions, like the complex lorentzian. This reviewer has been playing with these functions and even used the same technique to account for the magnetic scattering signal over 1 keV (not published) in an antiferromagnetic compound. In short, the amplitude difference between the resonant and the non resonance scattering locates the position of the minimum of the intensity or the disappearance (or not) of the diffracted signal. The overall sign of the non-resonant terms, equal or different than that of the resonant terms, will affect the location of the minimum; either before the edge or above. By this means one have a very powerful way to detect very small variation of the intensity of the resonant and non resonant signals, either as a function of temperature via the Debye-Waller (Figure 1) or as a function of the distortion induced by an electric field (Figures 3 and 5), and hence of the atomic positions. This interference property is what the authors have used in this paper.

- Following the previous point, it should be nice to enlarge the Supplementary Material chapter with some of the details that I have discussed.

- The part entitled "Sensitivity to thermal motion", although crucial to see how tiny variations of the intensity (in this case the Debye-Waller factor) affect the interference profile, is not part of the story that the authors want to make in this paper. I suggest to put it into the Supplementary Information chapter.

- As above, the paragraph "Application to other structures" should better fit in the Supplementary Information.

- In their analysis authors have assumed that Sr ions do not move. In the space group $P4mm$ Wyckoff site for Sr (1a) is $(0,0,z)$, and therefore some movement is allowed. Can you please discuss this choice ?

The data are convincing, the technique is very powerful and I feel the information conveyed in this manuscript is important and deserves to be published in Nature Communications. However the manuscript needs to be tidy up and the questions raised above need to be addressed in a proper way before accepting the manuscript for publication.

As the authors are trying to convince the scientific community of a new method, rather than simply presenting an interesting new result I think they need to present a much clearer explanation of the method along with a real assessment of the benefits of their method before this paper can be published.

Authors' response:

The referee points out an important issue which was also noticed by all other referees: Though all underlying physics have been presented in the methods part, there is a lack of description or illustration. This also seems to be necessary to understand the differences between the presented method and existing resonant diffraction techniques for structure refinement. In the revised version, this description is now given in the additional section **I.1 RXD near destructive interference**. The comment further addresses the question of unique experimental applicability and comparison to single energy data. Resolution of regular XRD structure determination is in general based on a large number of reflections with preferably high momentum transfer. In the presented scientific case, the MFP phase reversibly forms exclusively within a surface region of the SrTiO₃ single crystal. It cannot be separated and regular XRD on the phase is encumbered by a very limited amount of accessible reflections. With the new method, this lack of information can be compensated by the energy dependence of RXD. In addition, precision of atomic positions is highly increased up to the picometer level by maximizing the contrast of the signal, i.e. fitting the resonantly suppressed energy regions of the diffracted intensity for a set of reflection that weight the unknown structure parameters differently. To meet the referee's expectations, an addition in the discussion now addresses unique applicability and further compares the new method to other techniques.

Actions taken:

We edited the text accordingly improving the discussion in respect to unique benefits of the new method.

Reviewer #2 (Remarks to the Author):

2.1) *This paper reports a resonant X-ray study of the formation of a new crystalline (MFP) phase of SrTiO₃ under electric field with the goal being too clearly expose the atomic displacement in this restricted geometry system.*

From the perspective of the science studied in this work: I am not an expert in resonant X-ray measurements, SrTiO₃ or piezoelectricity, but I don't think solving this problem of the phase of SrTiO₃ is a burning question -- a quick check of the literature on the MFP phase indicates that previous work on this is not particularly highly cited and does not seem to have taken off much beyond the immediate efforts of a subset of the present authors.

From the perspective of a measurement achievement: The experiments themselves seemed compelling, although the details were hard to follow. There is a huge jump in depth of language/discussion/etc moving from the Introduction to the first Section of the paper. While it seems that all technical information is there, the relationship between different aspects and the flow does not seem optimal. Just to take one example: the authors state that a proper understand/quantification of the Debye-Waller factor on the resonance for bulk SrTiO₃ is necessary to expose the changes in the film geometries, but the rationale for that is not stated in adequately simple physical terms. A similar comment could be made about the issue of attenuation.

Authors' response:

This is a very helpful response and it is completely correct that in the first version we restricted ourselves to briefly present the underlying physics, which is in fact not new, before reporting the new methodology and structural results. It is true that the direct relation between the strong features in the resonant diffraction curves and the atomic positions was not clear, since contributions of different atoms, dynamic as well as static, are strongly entangled. Nevertheless, for the reduced case of a single displaced atom it is possible to easily depict these dependencies. To understand the influence of dynamic displacements and to improve the flow, these details are now described in an additional section (**I.1 RXD near destructive interference**).

In respect to absorption, we now state in the text that it has no qualitative impact on the main features but that a careful correction is necessary to obtain a reliable fit at the higher energies.

Regarding the importance of the structure of the MFP phase, we unfortunately cannot fully agree with the reviewer. This is for several reasons which are now stressed in the revised version of the manuscript.

First of all, as strontium titanate is extremely often used in layered heterostructures and the possible structural effects caused by a high electric field should play a big role for electronic applications. In fact, the article that reported the existence of the MFP phase for the first time (Hanzig, 2013) is now cited many times including 5 citations from papers of the Nature Publishing Group:

Nature Communications 5, Article number: 5554 (2014)

Nature Communications 6, Article number: 8585 (2015)

Scientific Reports 4, Article number: 3657 (2014)

Scientific Reports 5, Article number: 14576 (2015)

Scientific Reports 6, Article number: 22418 (2016)

Apart from that, we are not aware that such structural transformations have been triggered by a DC electric field in any other material before and the microscopic mechanisms behind formation and collapse of the MFP phase as well as its dependence on strain and sample thickness are not entirely understood yet.

Together with the recently published piezoelectric coefficient, the presented structural data adds to the understanding of the role of the substrate for the MFP layer morphology and its dynamics as well as its interplay with polarization, electrostriction and layer thickness. A corresponding paragraph was added in the discussion to highlight this relation.

Actions taken:

To understand the influence of dynamic displacements and to improve the flow we added an additional section (**I.1 RXD near destructive interference**). In respect to absorption, we now state in the text that it has no qualitative impact on the main features but that a careful correction is necessary to obtain a reliable fit at the higher energies.

2.2) Perhaps more importantly, I would have thought that for demonstrating the merit of a new method (or a variation), a more conventional system which is well (if not fully) understood would be more suitable than considering a problem (the MFP phase) that is open.

Authors' response:

We think that the chosen scientific case of a layered phase perfectly fits the aim to demonstrate the analytic power of the new methodology to acquire structural data which would not be accessible by means of conventional structure analysis methods. The reported structure solution for the recently discovered ambient temperature phase of one of the most representative oxides in our opinion significantly adds to the relevance of the work. Further, cubic strontium titanate is a well-known system. Therefore, the high precision determination of Debye Waller factors which are in excellent agreement with less precise literature values, already demonstrates the reliability of the new method.

2.3) Finally, considering other Nature Communication papers I have read, I would argue that the present paper is not written to the standard of readability for a sufficiently broad audience that I would expect for Nature Communications.

Given what I perceive to be a niche problem (at the present time) and the overwhelmingly technical tone of the paper, I would recommend that the authors submit their work in a more specialized journal such as Physical Review B. I therefore recommend that this paper be rejected from Nature Communications.

Authors' response:

We acknowledge that the first version of the manuscript was not easy to follow for readers not acquainted with Resonant X-ray Scattering since a general introduction leading to the new principles of the methodology had been omitted. In the revised version this is now included in a much clearer step by step outline, followed by a demonstration of easy dependencies and independently discussed influences of dynamic and static displacements on the signal before presenting the new structural results and applicability considerations for the new method.

Actions taken:

The outline of the manuscript was revised and a new introductory section has been added. Many paragraphs have been rephrased accordingly as well.

Reviewer #3 (Remarks to the Author):

3.1) *The manuscript describes a resonant X-ray technique to extract polar displacements in the MFP phase of SrTiO₃. The formation of the MFP was described in earlier publications.*

The technique is relatively new: others have used similar techniques to extract displacements in perovskites [Phys. Rev. Lett. 105, 217601 (2010); J. Appl. Crystallogr. 37, 193 (2004) -- these should be cited], but did not analyze the diffraction anomalous fine structure energy spectrum. The method used by the authors appears to be robust, with appropriate detail given to error analysis, and the conclusions appear valid.

Authors' response:

We thank the referee for giving credit to the novelty of the method and also for pointing out the reference to other approaches for displacement analysis, which we now acknowledge in the revised version. However, we have to emphasize that other attempts to extract structural information reported in literature are subject to substantial limitations compared to our development. These differences are now highlighted in a new paragraph in the discussion (see below).

Actions taken:

We added a new paragraph to the discussion: "There are similar attempts to extract structural information reported in literature, all subject to substantial limitations compared to our development. To avoid ambiguities in the refinement, fixed structural models were presupposed or only one structural parameter was assumed to be unknown^{38;39}. Often, the analysis required inverting ("switching") the polarity of the structure^{40;41} with an electric field in order to measure the Friedel pair contrast. This is not always feasible or may alter the material as in the present case. Furthermore, in all cases the impact of Debye-Waller factors was neglected and the obtained resolution was much lower than in our approach which exploits destructive interference for the first time."

3.2) *However, this reviewer feels that the technique of "resonantly suppressed X-ray diffraction" will be of very limited interest, due to the need for a synchrotron, the time necessary to measure the data (and for analysis), the specificity of the method (small atomic displacements), and the fact that other crystallographic techniques (e.g., phase retrieval techniques [Nat. Mater. 2, 99 (2002); J. Phys. Condens. Matter 20, 445006 (2008); etc.]) can be used to extract similar position information, although with not quite the resolution, but over the full thickness of the unknown layer. Publication in a more specialized journal would be appropriate.*

Authors' response:

Indeed the method cannot be applied routinely and energy scans are still bound to synchrotron radiation sources. There are other methods that can yield atomic displacements to a certain degree for the majority of scientific cases.

Nevertheless, for many cases resolution is important e.g. for subtle symmetry changes during phase transformations at morphotropic phase boundaries. Furthermore the presented method excels in cases where samples are heterogeneous, to separate mixed phase systems, buried layers or strained regions in reciprocal space, and does not average over displacements within the whole layer.

Regarding the time consuming data acquisition and analysis we are confident that there is a lot of room for future improvements.

Reviewer #4 (Remarks to the Author):

4.1) This is an interesting paper that reports on picometer atomic displacements as a result of the application of an electric field on SrTiO₃. The goal and interest of this study has been clearly identified in the abstract and in the introduction, and I do not want to copy and paste the overall originality of this work, which I fully agree with. It is important to characterise these distortions, the method that the authors have used is well suited (see more below) and this reviewer believes that the manuscript should be published in Nature Communications, albeit not in its present form.

Authors' response:

We are grateful for this acknowledgement in support of the new methodology and a publication in Nature Communications.

4.2) Going further into the details of the manuscript, I have the following remarks: The wording of the title "...by resonantly suppressed X-ray diffraction" is not appropriate. This sentence is not correct and turns out to be meaningless without reading the paper, which is not the primary goal of a title. Authors should seek for a better title.

Authors' response:

We thank the referee for pointing out the need to substantiate the title in making it clearer.

Actions taken:

The new title which we think adequate for the manuscript is "Picometer polar displacements in strontium titanate determined by a new approach of resonant x-ray diffraction."

4.3) I disagree with the authors' statement that they "have developed a novel method of X-ray structure analysis". Fitting the energy profiles to account for crystallographic information is currently done and there are examples in the literature. Reference 10 includes some "old" examples.

Authors' response:

We agree that “novel method of X-ray structure analysis” might be a bit overstated.

Actions taken:

We rewrote the entire introduction and avoided this formulation. Instead, we now claim to use “a new way/approach of resonant x-ray diffraction for structure analysis”.

4.4) The authors have been very vague about the details of their method and the basic underlying idea is missing. Authors should capture in simple words the main features of their methodology. I agree that the method is embedded in Equation 3. However it is not convincing and the reader has to take authors' word for granted. The characteristic profiles measured and shown in Figures 1, 3, 5 and S1 result from the interference of a rapidly varying complex function, the resonant scattering factor, with a smoothly decreasing non resonant (Thomson) scattering. This kind of profiles stands not only for tabulated f' and f'' but for a variety of complex functions, like the complex lorentzian. This reviewer has been playing with these functions and even used the same technique to account for the magnetic scattering signal over 1 keV (not published) in an antiferromagnetic compound. In short, the amplitude difference between the resonant and the non-resonance scattering locates the position of the minimum of the intensity or the disappearance (or not) of the diffracted signal. The overall sign of the non-resonant terms, equal or different than that of the resonant terms, will affect the location of the minimum; either before the edge or above. By this means one have a very powerful way to detect very small variation of the intensity of the resonant and non-resonant signals, either as a function of temperature via the Debye-Waller (Figure 1) or as a function of the distortion induced by an electric field (Figures 3 and 5), and hence of the atomic positions. This interference property is what the authors have used in this paper. Following the previous point, it should be nice to enlarge the Supplementary Material chapter with some of the details that I have discussed.

Authors' response:

The reviewer is correct that the original version lacked a description of the new approach which is why we extended the introduction to the methodology. The view and explanations of the reviewer have been very helpful to formulate this section.

Actions taken:

We added the section (**I.1 RXD near destructive interference**) to illustrate the underlying idea.

4.6) The part entitled "Sensitivity to thermal motion", although crucial to see how tiny variations of the intensity (in this case the Debye-Waller factor) affect the interference profile, is not part of the story that the authors want to make in this paper. I suggest to put it into the Supplementary Information chapter.

Authors' response:

The characterization of the thermal displacement is important for several reasons. The drastic change in energy-dependence demonstrates the sensitivity of the chosen reflections. What is more, determining the dynamic displacement parameters of the atoms (ADPs) serves as a test case to verify the reliability of the method. This was also demanded by other reviewers. Finally, only the correct values for the ADPs allow a correct refinement of static atomic displacement in the MFP phase.

Actions taken:

The title of the section was changed to "Characterization of dynamic, thermal displacement".

4.6) As above, the paragraph "Application to other structures" should better fit in the Supplementary Information.

Authors' response:

The authors agree.

Actions taken:

The section "Application to other structures" was moved to the supplementary information.

4.7) In their analysis authors have assumed that Sr ions do not move. In the space group $P4mm$ Wyckoff site for Sr (1a) is $(0,0,z)$, and therefore some movement is allowed. Can you please discuss this choice?

Authors' response:

We can arbitrarily choose the Sr z position as the origin of the unit cell. All other positions are given with respect to this fixed origin.

Actions taken:

We clarified this in the caption of table 2.

4.8) The data are convincing, the technique is very powerful and I feel the information conveyed in this manuscript is important and deserves to be published in Nature Communications. However the manuscript needs to be tidy up and the questions raised above need to be addressed in a proper way before accepting the manuscript for publication.

Authors' response:

We thank the referee for supporting this work to be published in Nature Communications and for his critic and helpful comments to improve the manuscript accordingly.

Reviewers' comments:

Reviewer #1 (Remarks to the Author):

I found the new version of the paper to be a dramatic improvement over the earlier one. The authors have clearly taken my comments, and those of the other referee's, about the need for a more clear presentation and discussion of their method seriously and spent considerable time in addressing this concern.

As such, I am now happy to be able to recommend this paper for publication in Nature Communications.

Reviewer #4 (Remarks to the Author):

This reviewer is satisfied with the way the authors have amended their manuscript. This final version turns out to be a little lengthy but it reads very well. I have no further comments and I find this manuscript suitable for publication in Nature Communications.

Reviewer #5 (Remarks to the Author):

This manuscript presents on a MFP of STO by polarizing a single crystal with a large electric field. The lattice spacing/symmetry change and related fine structural changes were measured using an x-ray resonance diffraction technique. The main points of the paper are of two folds: (i) lattice expansion of STO is novel or important and (ii) the x-ray resonance diffraction technique used here is new and sensitive to precision measurements enabling to discover or determine the MFP.

Although the measurements and presentation are interesting overall, I am unsure of the validity of the data analyses and conclusion. The equations presented in the method section do not provide the equation used in the fits or simulations in their data analysis. At least, it does not so clearly enough for me to follow and check the validity. Where is the equation used to produce Figure 1? Where does the sensitivity to displacement and square deviation come from?

The measurements were done on so-called anti-Bragg in their measurements. Under anti-Bragg condition, unfortunately, the intensity does not go to zero (this is common misconception in kinematic approximation). That is why crystal truncation rods are so powerful in studying surfaces. There have been quite of discussion and use of this anti-bragg over decades. Resonance scattering on the anti-Bragg has also done, mainly surface scattering point of view. The anti-bragg is surface or near surface sensitive, therefore resonance scattering at the anti-bragg is also surface or near surface sensitive. The measurements presented here might all be explained by some near surface lattice expansion, not that of the bulk. Ti electrode does not contribute to the anti-bragg because Ti electrode atoms are not registered with STO lattices.

I wonder if it is novel or new that electric polarization induces oxygen deficiency and related chemical expansion on perovskites. The whole solid oxide fuel cell community has been studying the oxygen deficiencies, related conductivity, and chemical expansion of various perovskites. STO is not very much studied in that regard simply because STO is a lousy ionic conductor. Also, the precision in lattice constant measurement using synchrotron x-rays routinely goes some picometers without

having to work too hard or using the resonance. Therefore, the precision measurement is nothing new either. I just googled "LSCF lattice constant synchrotron" and find several related articles or slides on the topic.

Reviewer #6 (Remarks to the Author):

I have reviewed the reports of the other referees, the authors' response, and the revised manuscript. Overall, my opinion is that the authors have addressed the concerns raised by the reviewers. I have, however, a few areas of concern:

1. First, following up on the first reviewer's concern, which says "My most serious concern is that I do not feel that the novel method is presented in a very clear manner, and certainly not one that a non-specialist can follow." I agree with that assessment, even after the response to the reviews.

My specific concern in this area is that figure 1 and the associated discussion are extremely difficult to follow, especially for non-experts. I suspect that non-experts (e.g. readers wishing to learn about the crystallography of SrTiO₃ but who are themselves not experts in x-ray scattering) will not realize that f' and f'' are being plotted as a function of photon energy. Experts will not know which direction the energy increases, or what assumptions are being made about the variation of the structure factors. The discussion of the contributions of Sr Ti and O atoms and the inset diagrams are very difficult to follow and would in fact merit a separate sub-figure. What energy, for example, is the vector sum shown for?

2. I believe the paper would be significantly improved by adding brief descriptions of the fundamental basis of the approximations used in the paper.

2.a. The development of the MFP phase is presumably accompanied by a chemical change of the SrTiO₃. The authors should provide a brief quantitative justification of their use of the bulk atomic scattering factors for all phases. This is important because it is possible in principle that changes in f' and f'' due to chemical effects (e.g. as reported in absorption spectra of SrTiO₃ during crystallization in Frenkel et al PRL 99, 215502 (2007)) could produce changes in the resonant scattering. This seems to me to be unlikely, but should be explicitly discussed by the authors.

2.b. The authors account for a gradient in the structure of the MFP phase through the approximation that the scattering at each depth corresponds to a specific wavevector. This approximation is not valid if the MFP layer is thin or the strain gradient is very high. The authors should quantitatively evaluate the basis for this approximation and state it in the paper.

My opinion is that once these concerns are addressed the manuscript will be suitable for publication.

Reviewer #1:

I found the new version of the paper to be a dramatic improvement over the earlier one. The authors have clearly taken my comments, and those of the other referee's, about the need for a more clear presentation and discussion of their method seriously and spent considerable time in addressing this concern.

As such, I am now happy to be able to recommend this paper for publication in Nature Communications.

We are glad that the manuscript meets the expectations of reviewer #1. We thank the reviewer for his positive feedback and his recommendation to publish the work in Nature Communications.

Reviewer #4:

This reviewer is satisfied with the way the authors have amended their manuscript. This final version turns out to be a little lengthy but it reads very well. I have no further comments and I find this manuscript suitable for publication in Nature Communications.

We also thank reviewer #4 for the friendly support of the manuscript to be published in Nature Communications.

Reviewer #5:

“This manuscript presents on a MFP of STO by polarizing a single crystal with a large electric field. The lattice spacing/symmetry change and related fine structural changes were measured using an x-ray resonance diffraction technique. The main points of the paper are of two folds: (i) lattice expansion of STO is novel or important and (ii) the x-ray resonance diffraction technique used here is new and sensitive to precision measurements enabling to discover or determine the MFP.”

“Although the measurements and presentation are interesting overall, I am unsure of the validity of the data analyses and conclusion. The equations presented in the method section do not provide the equation used in the fits or simulations in their data analysis. At least, it does not so clearly enough for me to follow and check the validity. Where is the equation used to produce Figure 1? Where does the sensitivity to displacement and square deviation come from?”

We are grateful to the referee for the careful examination of the physical validity of the method. We have to emphasize that the physics and all equations that are needed to reproduce the calculations are indeed contained in the methods part. The equation to reproduce the fits, Fig. 1 and the sensitivity to displacement are (1), (3) and (5), respectively (numbering according to the new version).

Nevertheless, the referee correctly pointed out that these equations were not referenced sufficiently to provide a clear description. To take account of this criticism, we now refer to these at the necessary places in the section I.1 including the caption of Fig. 1. We also added details of the simulation and improved the explanation in the caption of Fig. 1 and the main text.

“The measurements were done on so-called anti-Bragg in their measurements. Under anti-Bragg condition, unfortunately, the intensity does not go to zero (this is common misconception in kinematic approximation). That is why crystal truncation rods are so powerful in studying surfaces. There have been quite of discussion and use of this anti-bragg over decades. Resonance scattering on the anti-Bragg has also done, mainly surface scattering point of view. The anti-bragg is surface or near surface sensitive, therefore resonance scattering at the anti-bragg is also surface or near surface sensitive. The measurements presented here might all be explained by some near surface lattice expansion, not that of the bulk. Ti electrode does not contribute to the anti-bragg because Ti electrode atoms are not registered with STO lattices.”

We honestly think that there has been a misconception about the experimental procedure which is due to a certain similarity of different methods. We regret that the technique was not presented clear enough in the former manuscript to avoid such kind of confusion.

First of all, we did not measure in anti-Bragg condition, but always at the exact Bragg condition. Using resonance effects to promote destructive interference at the Bragg condition might appear similar to de-tuning away from the Bragg condition by variation of the momentum transfer. But, in contrast to crystal truncation rods (CTR), we keep the momentum transfer constant and tune the scattering amplitudes by variation of energy. Furthermore, the surface sensitivity of anti-Bragg scattering was not an aim of our work. It is usually sensitive to only a few atomic layers while we observe a volume-effect, even if only in a layer of a few microns thickness.

In the revised version of the paper, we improved the explanation of our methodological approach, again in section I.1, in order to point out more explicitly the measurement strategy. Furthermore, we added a comparison with related X-ray techniques for structure determination (including crystal

truncation rods) in the revised Supplement (VI.2 Range of application) and discuss eventual interrelationships.

“I wonder if it is novel or new that electric polarization induces oxygen deficiency and related chemical expansion on perovskites. The whole solid oxide fuel cell community has been studying the oxygen deficiencies, related conductivity, and chemical expansion of various perovskites. STO is not very much studied in that regard simply because STO is a lousy ionic conductor.”

The referee raises an interesting point and is correct with the remark regarding solid oxide fuel cells (SOFCs). However, the presented case is very different. Not only were no oxygen vacancies introduced into the crystal but in fact the samples were pre-annealed in oxygen at 900°C to remove them. Thus, the investigated phase is based on the dynamics of the tiny fraction of the thermodynamically unavoidable remaining oxygen vacancies. From our latest studies we can conclude a maximum concentration limit of $10^{18}/\text{cm}^3$ [Hanzig2016 10.1088/0953-8984/28/22/225001]). Finally, it should be noted that the MFP-Phase is formed in the region of the crystal, where oxygen vacancies are further depleted due to the applied electric field [Hanzig2013 10.1103/PhysRevB.88.024104].

Electrocoloration experiments [Waser1990, 10.1111/j.1151-2916.1990.tb09810.x] prove this ionic transport and the vacancy accumulation at the electrode which lies on the opposite site of the rather thick sample. Therefore, we are certain that the MFP structure does not exhibit oxygen deficiency.

Comparing the indeed well-known stoichiometry dependence of the chemical expansion on dopant concentration (e.g. [Bishop14 10.1146/annurev-matsci-070813-113329]) it is clear that the MFP-Phase represents the case of the dopant-free limit, in which no chemical expansion is expected. A case in point is the oxygen vacancy enriched cathode prepared in the process of the electroformation which does not exhibit structural changes.

Furthermore, we would like to emphasize that the intended focus of the manuscript is the solution of the crystal structure of the MFP phase by means of a newly developed resonant diffraction technique (see below). From the viewpoint of materials science, the prospect of transferring the developed method to other material systems as discussed in the revised supplement section VI.2 might be more interesting.

“Also, the precision in lattice constant measurement using synchrotron x-rays routinely goes some picometers without having to work too hard or using the resonance. Therefore, the precision measurement is nothing new either. I just googled “LSCF lattice constant synchrotron” and find several related articles or slides on the topic.”

Regarding this remark of the referee, we would like to point out that unit cell determination providing cell parameters only and structure solution providing atomic positions as well are related but different tasks. On top of that, the solution of two stoichiometrically identical, structurally similar and intergrown crystal structures is a much more complicated task, which cannot be achieved by any conventional structure determination approach. Therefore we have to ask the referee to reconsider this point of view .

In addition to the revised version of the manuscript, we also enclose a file with highlighted changes in the text. We are very grateful to the referee for the very constructive remarks and hope that the revised version of the manuscript will meet the requirements.

Reviewer #6:

I have reviewed the reports of the other referees, the authors' response, and the revised manuscript. Overall, my opinion is that the authors have addressed the concerns raised by the reviewers. I have, however, a few areas of concern:

"1. First, following up on the first reviewer's concern, which says "My most serious concern is that I do not feel that the novel method is presented in a very clear manner, and certainly not one that a non-specialist can follow." I agree with that assessment, even after the response to the reviews.

My specific concern in this area is that figure 1 and the associated discussion are extremely difficult to follow, especially for non-experts. I suspect that non-experts (e.g. readers wishing to learn about the crystallography of SrTiO₃ but who are themselves not experts in x-ray scattering) will not realize that f' and f'' are being plotted as a function of photon energy. Experts will not know which direction the energy increases, or what assumptions are being made about the variation of the structure factors. The discussion of the contributions of Sr Ti and O atoms and the inset diagrams are very difficult to follow and would in fact merit a separate sub-figure. What energy, for example, is the vector sum shown for?"

We thank the referee for addressing this lack of comprehensibility regarding Fig. 1. The Figure was introduced to address concerns formerly expressed by the reviewers #1 and #4 about the explanation of the technique and it is correct that some background information was missing.

To take account of the concerns, we added some more information to the figure, for instance about the direction of the X-ray energy variation and the different structural influences. We also extended the caption of Fig.1 and the surrounding text, used more explicit terms and added details about the simulation. We hope that the new version is clear enough without an additional figure and are looking forward to the reviewers opinion.

2. I believe the paper would be significantly improved by adding brief descriptions of the fundamental basis of the approximations used in the paper.

2.a. The development of the MFP phase is presumably accompanied by a chemical change of the SrTiO₃. The authors should provide a brief quantitative justification of their use of the bulk atomic scattering factors for all phases. This is important because it is possible in principle that changes in f' and f'' due to chemical effects (e.g. as reported in absorption spectra of SrTiO₃ during crystallization in Frenkel et al PRL 99, 215502 (2007)) could produce changes in the resonant scattering. This seems to me to be unlikely, but should be explicitly discussed by the authors.

The reviewer puts a valid question that is now discussed in the manuscript. We expect the chemical effects to the fine structure in f' and f'' to be negligible. One reason is the sample preparation starting from stoichiometric single crystalline strontium titanate that is annealed in oxygen at 900°C. The electroformation process producing the MFP-Phase then accumulates the remaining oxygen vacancies on the other side of the sample which is not probed by the X-rays. Thus, the part of the crystal under investigation is as close as possible to the ideal stoichiometry Sr₁Ti₁O₃ and chemical effects on f' and f'' are not to be expected.

What remains are changes in the fine structure due to geometrical relaxation of the atoms and the electron density. To discuss these, we now show XANES measurements at the Ti-K edge, which show only marginal changes during the MFP formation process. Besides, we add a further discussion of the related influences on the measurements in the supplement of the manuscript (section VI.1).

2.b. The authors account for a gradient in the structure of the MFP phase through the approximation that the scattering at each depth corresponds to a specific wavevector. This approximation is not valid if the MFP layer is thin or the strain gradient is very high. The authors should quantitatively evaluate the basis for this approximation and state it in the paper.

This is a very important remark on an issue which has not been addressed accordingly in the previous version of the manuscript. To meet the expectations of the referee we now amended a paragraph in section I.4, providing a quantitative analysis of the strain gradient and the respective conclusions on the strain resolution in the experiment:

One should note that the model of a strain gradient predicts that there is no region of constant strain with a finite width. From the experimental conditions, one can estimate how thick the layers are that contribute to a certain position in L-scans of reciprocal (Q) space. This assessment is based on the value of the strain gradient. In an exponential strain profile model for our case, we obtained an upper limit for the strain gradient of $1.5\mu\text{m}^{-1}$ (near the surface) corresponding to a value of $5.9\text{E-}7$ per unit cell. For the 007 reflection, this would result in a $\pi/2$ phase shift after 350 unit cells and therefore in a thickness of 140 nm of the contributing layer.

Based on this thickness, the calculated peak width for the 007 reflection is $\Delta L \sim 0.0035$. This estimation has been done for the position of highest strain. For lower strain, the broadening of Bragg peaks will decrease and the depth resolution of the measurement will improve. Therefore, the sampling we used in this work was always below the limits of resolution.

My opinion is that once these concerns are addressed the manuscript will be suitable for publication.

In addition to the revised version of the manuscript, we also enclose a file with highlighted changes in the text.

We are very grateful to the referee for the very constructive remarks and hope that the revised version of the manuscript will meet the requirements.

Reviewers' comments:

Reviewer #5 (Remarks to the Author):

In the light of many reviewers involved, I do not want to further delay of this publication, especially many seem OK with the manuscript. I am willing to let this published after the author review once more the following comments about the anti-Bragg condition since the authors may not be an expert in this regard.

The (HKL) with $H+K+L=\text{odd}$ is a Bragg condition as defined in the cubic unit cell. Its intensity is precisely zero in cubic symmetry. Or it is weak in case the symmetry has been broken. Since the resonance condition of metal atoms can break the symmetry due to the imaginary part, it can be sensitive to the oxygen positions and vibration. (BTW, this is not clearly shown as I pointed out in the earlier report) However, these Bragg points are also anti-Bragg points from surface scattering point of view (Explanation of why is beyond the scope of this report), and there will be the contribution from the metal atoms (with or without resonance conditions met) due to the broken symmetry nature of the surface. I suspect the surface is really smooth and well defined since it is annealed at 900C. I would say then the surface contribution will be fairly strong. At least there is no clear explanation in the theory or experiment whether the CTR is included or avoided.

Reviewer #6 (Remarks to the Author):

The authors have addressed my previous concerns adequately. My opinion is that the manuscript is now suitable for publication.

Reviewer #5 (Remarks to the Author):

In the light of many reviewers involved, I do not want to further delay of this publication, especially many seem OK with the manuscript. I am willing to let this published after the author review once more the following comments about the anti-Bragg condition since the authors may not be an expert in this regard.

The (HKL) with $H+K+L=odd$ is a Bragg condition as defined in the cubic unit cell. Its intensity is precisely zero in cubic symmetry. Or it is weak in case the symmetry has been broken. Since the resonance condition of metal atoms can break the symmetry due to the imaginary part, it can be sensitive to the oxygen positions and vibration. (BTW, this is not clearly shown as I pointed out in the earlier report) However, these Bragg points are also anti-Bragg points from surface scattering point of view (Explanation of why is beyond the scope of this report), and there will be the contribution from the metal atoms (with or without resonance conditions met) due to the broken symmetry nature of the surface. I suspect the surface is really smooth and well defined since it is annealed at 900C. I would say then the surface contribution will be fairly strong. At least there is no clear explanation in the theory or experiment whether the CTR is included or avoided.

Answer from the authors:

We appreciate the view on the diffraction process that the reviewer provided from a different perspective. Indeed the set of reflections which we studied correspond to anti-Bragg points for a body-centered cell. However, in our case, this symmetry is already broken since there are different cations situated at the center (Ti) and the corners (Sr) of the unit cell. This difference in terms of the X-ray characteristics become very small at the Sr edge and for the X-rays, the cell seems almost body-centered. This is why we are again very sensitive to small displacements, as it was described in Ch. 1.1.

The CTR contribution would cause a constant offset of the energy scans we analysed. Even at energies where the Bragg intensity is minimal (Sr and Ti scatter very similarly) we still see a pronounced Bragg maximum in the L-scans showing that the CTR is a minor contribution in our case. We therefore neglected the CTR contribution.

Actions taken:

We adapted Supplementary Note 2, and discuss the contribution of CTR in a paragraph there. This paragraph now reads:

``

The analysis of crystal truncation rods (CTRs) in reciprocal space bears similarity to the presented method, since in both cases the substantial variations in the signal are observed at the slopes, away from intensity maxima. In the body centered unit cell, a selection rule exists for the set of Bragg reflections that was in focus for our analysis (one odd miller index). This means they would be precisely zero if the two metal cations (Sr, Ti) would have the same scattering amplitude. This can also be referred to as ``anti-Bragg'' condition where surface contributions (CTR) to the scattered intensity become important. When we use the energy dependence to reduce the Sr scattering amplitude, we come close to this situation and, in principle, CTR contributions may play a role. However, the fundamental difference is that we maintain the Bragg condition for each selected crystalline region in our case of resonantly suppressed diffraction (RSD): The enormous sensitivity to positional changes of atoms is achieved through the analysis of energy dependencies near destructive interference effects while we reside at a fixed position of reciprocal space. Therefore, the CTR would only result in a constant offset of the whole spectrum. In our case it was it was not necessary to take such contribution into account, because even though the Bragg intensity at the minimum is reduced by several orders of magnitude, it is still much stronger than the surface scattering. This assessment can be made based on the clear appearance of a Bragg maximum from cubic even at the energy of destructive interference (see black crosses in Fig 7).''

REVIEWERS' COMMENTS:

Reviewer #5 (Remarks to the Author):

My concerns and comments were properly addressed. I will be happy to see the manuscript published.